# Seroprevalence Study of Conserved Enterotoxigenic *Escherichia coli* Antigens in Globally Diverse Populations

**DOI:** 10.3390/microorganisms11092221

**Published:** 2023-08-31

**Authors:** Frederick Matthew Kuhlmann, Vadim Grigura, Timothy J. Vickers, Michael G. Prouty, Lora L. Iannotti, Sherlie Jean Louis Dulience, James M. Fleckenstein

**Affiliations:** 1Division of Infectious Diseases, Department of Medicine, Washington University School of Medicine in Saint Louis, Saint Louis, MO 63110, USA; fmkuhlmannwu@gmail.com (F.M.K.); vadim.grigura@wustl.edu (V.G.); tjvickers@wustl.edu (T.J.V.); 2U.S. Naval Medical Research Unit No. 6—NAMRU 6, Lima 15001, Peru; michael.g.prouty2.mil@mail.mil; 3Institute for Public Health, Brown School, Washington University in Saint Louis, Saint Louis, MO 63110, USA; liannotti@wustl.edu (L.L.I.); sherliejeanlouis@yahoo.fr (S.J.L.D.); 4Medicine Service, Infectious Diseases, Saint Louis VA Health Care System, St. Louis, MO 63110, USA

**Keywords:** enterotoxigenic *Escherichia coli*, diarrhea, vaccine, antigens, surface, antibodies

## Abstract

Enterotoxigenic Escherichia coli (ETEC) are common causes of infectious diarrhea among young children of low-and middle-income countries (LMICs) and travelers to these regions. Despite their significant contributions to the morbidity and mortality associated with childhood and traveler’s diarrhea, no licensed vaccines are available. Current vaccine strategies may benefit from the inclusion of additional conserved antigens, which may contribute to broader coverage and enhanced efficacy, given their key roles in facilitating intestinal colonization and effective enterotoxin delivery. EatA and EtpA are widely conserved in diverse populations of ETEC, but their immunogenicity has only been studied in controlled human infection models and a population of children in Bangladesh. Here, we compared serologic responses to EatA, EtpA and heat-labile toxin in populations from endemic regions including Haitian children and subjects residing in Egypt, Cameroon, and Peru to US children and adults where ETEC infections are sporadic. We observed elevated IgG and IgA responses in individuals from endemic regions to each of the antigens studied. In a cohort of Haitian children, we observed increased immune responses following exposure to each of the profiled antigens. These findings reflect the wide distribution of ETEC infections across multiple endemic regions and support further evaluation of EatA and EtpA as candidate ETEC vaccine antigens.

## 1. Introduction

Despite overall reductions in mortality from infectious diarrhea, the incidence of diarrheal disease due to bacterial pathogens and associated morbidity among vulnerable children remain high [1,2,3]. Enterotoxigenic *Escherichia coli* (ETEC) are among the most common contributors to the diarrheal disease burden [4]. Acute presentations of ETEC infections can range from mild diarrhea to severe cholera-like watery diarrhea [5] and death. In contrast to *Vibrio cholerae*, ETEC are genetically heterogeneous organisms belonging to multiple serotypes [6] that are often associated with repeated episodes [7] of symptomatic disease in young children of low-middle income countries (LMICs). Repeated diarrheal illnesses from ETEC and other enteropathogens are associated with intestinal damage and enteric dysfunction leading to malabsorption, impaired growth and impaired cognitive development [8,9], negatively impacting human capital in low-income regions.

Hundreds of millions of ETEC infections are estimated to occur each year among infants and children in LMICs [10], as well as among travelers to ETEC endemic areas lacking necessities of clean water and basic sanitation [11]. Therefore, vaccine development remains a high priority. ETEC infections drive empiric antibiotic use, potentially promoting the development of antimicrobial resistance [12,13]. Fortunately, as infections decline significantly after the age of two years in endemic regions, natural infections ultimately appear to offer protection against symptomatic illness [7], suggesting that an ETEC vaccine is feasible, and several ETEC vaccine candidates are currently being evaluated [14].

ETEC share in their capacity to elaborate heat-labile (LT) and/or heat-stable (ST) enterotoxins [15]. Their diversity arises from surface-expressed antigens known as colonization factors (CFs). ETEC vaccines in development to date have principally targeted the most common colonization factors (CFs). However, the identification of more than 30 antigenically distinct target antigens has prompted investigation of other conserved antigens to complement existing approaches [16,17].

Two candidate antigens, the EatA mucinase [18,19], and the EtpA adhesin [20] have emerged in the search for novel ETEC virulence factors and might contribute to development of a broadly protective vaccine. Both proteins are specific to the ETEC pathovar and are highly conserved among a geographically diverse collection of isolates [21]. Both antigens are expressed by multiple strains linked to severe cholera-like illness [5,22,23], suggesting that these play important roles in disease. Likewise, in studies of a birth cohort of young Bangladeshi children, both antigens were strongly associated with symptomatic ETEC infections, and antibodies against either antigen appear to be associated with protection from symptomatic illness [24].

Details regarding structure and function of these secreted proteins are beginning to provide important insights into the molecular pathogenesis of ETEC, which can inform their use as vaccine candidates. EatA, a member of the serine protease autotransporter of *Enterobactericiae* (SPATE) family of molecules [19], specifically degrades MUC2 secreted by goblet cells [18] to facilitate the intimate interactions between ETEC and target intestinal epithelial cells required for efficient toxin delivery [25]. EtpA is a high-molecular-weight extracellular glycoprotein secreted by a two-partner secretion (TPS) system encoded by the *etpBAC* locus [26]. EtpA forms a molecular bridge [20] between the bacteria and N-acetylgalactosamine (GalNAc) target glycans abundant in intestinal mucosa [27] to accelerate delivery of both LT [28] and ST [29]. Notably, GalNAc is also the terminal sugar residue on human A blood group glycans present both on erythrocytes and intestinal epithelia. Specific interactions between EtpA and human A blood group glycans are thought to explain the increased diarrheal severity observed following ETEC infection among human volunteers [30] and young children [7] belonging to the A blood group.

To date, molecular epidemiology studies have suggested that these antigens may be broadly conserved across a diverse population of strains [21,31]. Still, additional data are needed to examine their distribution and immunogenicity among different populations at risk of ETEC infection. As seroprevalence estimates likely reflect the distribution of ETEC expressing these antigens, data from diverse populations could provide information regarding the degree to which the EtpA and EatA, alone and together, are present in different populations and guide antigen selection for vaccine development [32]. Therefore, we sought to expand our examination of serologic responses to these antigens to other ETEC endemic regions [24].

To advance our understanding of host responses to EatA and EtpA, we analyzed a convenience sample of serum and plasma available from diverse global settings using an optimized ELISA assay. We hypothesized that antibody responses to these antigens would be elevated in regions of endemicity. The analyses presented here corroborate our earlier molecular epidemiology studies suggesting that ETEC producing EtpA and EatA are widely distributed and that these antigens are immunogenic. These observations reenforce support for the further evaluation of EatA and EtpA as candidate ETEC vaccine antigens.

## 2. Materials and Methods

### 2.1. Recombinant Antigen Production

Polyhistidine-tagged recombinant secreted EtpA and EatA passenger proteins were purified from culture supernatants by metal affinity chromatography (HisTrap, Pharmacia) as described previously [33], while 8x-His tagged EatA passenger domain was purified from culture supernatant of strain Top10 (pTV001, jf3227) (Appendix A) and recombinant polyhistidine-tagged EtpA purified as previously described from culture supernatants of Top10 (pJL017/pJL0130, jf1696) following induction with 0.005% arabinose. Recombinant proteins were then examined by Coomassie staining to verify purity. Western immunoblotting was performed using human serum (1:4000 dilution) and anti-human IgG (dilution 1:1000) to assess the specificity of the immune response.

### 2.2. Human Serum Samples

Archived blood samples previously collected in 2016 from Cameroon [34] or circa 1997 from Egypt (Qalubiya Governorate) ([35]) had been obtained during the course of field studies on filarial infections. Samples from Cameroon had been preserved as dried blood spots on HemaSpot HF cards (Spot On Sciences, San Francisco, CA, USA), and had been stored in the dark at room temperature. One blade from the HemaSpot cards was submerged in a microfuge tube using 300 μL of PBS with 0.5% bovine serum albumin. The blade was kept at room temperature overnight, then centrifuged at 2500× *g* for 7 min and the supernatant was removed and stored. This suspension was estimated to equate to approximately a 1:200 dilution based on similar studies with dried blood spots (see Appendix A) [36].

Serum samples from Egypt previously stored at −20 °C were thawed and aliquoted for antibody detection. All participants were over 10 years of age. Deidentified serum samples collected during routine clinical care and that would have otherwise been discarded were obtained from St. Louis Children’s hospital (children < 2 years of age) or Barnes Hospital (adults aged 30–50 years of age). Deidentified samples from Peru included adults enrolled in a dengue seroprevalence study and similarly aged with no corresponding metadata. Samples from Haitian children (6 months to 3 years of age) were obtained from a separate pilot study [37] in which both fecal and plasma samples were collected one month apart.

### 2.3. ELISA

Additional information regarding assay development and optimization is provided in the Appendix A. ELISA parameters were first optimized using murine samples as described in the Appendix A. ELISAs targeting the novel antigens EatA and EtpA, as well as LT, were subsequently optimized to detect antibody responses in human samples. EtpA and EatA ELISA plates were prepared by coating individual wells with 100 μL of the respective proteins in carbonate buffer (15 mM Na_2_CO_3_, 35 mM NaHCO_3_, 0.2 g/L NaN_3_, pH 9.6), at 1 μg/mL or 10 μg/mL for EtpA and EatA, respectively. Plates were then incubated overnight at 4 °C. LT ELISA plates were generated by incubating plates overnight at 4 °C, with 1 μg/mL GM1 ganglioside (Avanti, Birmingham, AL, USA: 860065P) in carbonate buffer, 100 µL/well. The following day, plates were washed and then incubated with LT (courtesy of Dr. John Clements, Department of Microbiology and Immunology, Tulane University School of Medicine) 1 µg/mL overnight at 4 °C. Plates were washed 3 times in PBS with 0.05% Tween (PBST), dried for four hours at 37 °C, sealed, and stored in the dark at 4 °C for up to 3 months. When ready for use, plates were blocked with 1% BSA in PBS in for one hour at 37 °C then washed in PBST × 3. Primary sera were added at specified dilutions (1:800 for all IgG assays, 1:400 for anti EtpA IgA assays, and 1:200 for anti-LT and EatA IgA assays) and incubated for one hour at 37 °C, then washed five times in Phosphate-buffered saline–0.5% Tween 20 (PBS-T). Optimal screening dilutions were determined by performing 2-fold serial dilutions of a subset of samples to identify a dilution where the majority of samples remained in the dynamic range of the kinetic ELISA assay [38] with minimal background in negative controls (children in the US). Plates were incubated for 45 min with 1:20,000 dilution of HRP-conjugated rabbit anti-human IgG (Jackson ImmunoResearch Laboratories, West Grove, PA, USA: 309-035-006) or HRP-conjugated rabbit anti-human IgA antibodies (Jackson ImmunoResearch: 309-035-011). After washing, plates were developed using 3,3′ 5,5′-tetramethylbenzidine (TMB, SeraCare, Gaitersburg, MD, USA: 5120-0074). ELISA readings were recorded kinetically on a plate reader (BioTek, Santa Clara, CA, USA) and analyzed using Gen5 software (v2.00.18) as before [24,38]. Results were log_10_ transformed to normalize the distribution. Additional details on use of dried blood spots (DBS) are provided in Appendix A.

To assess avidity, freshly prepared 6 M urea in PBS was added after incubation with primary serum. Plates were incubated for 10 min at 37 °C, washed 3 times in PBS, followed by incubation with secondary antibodies as above. The ratio of Vmax in urea-treated vs. untreated wells (multiplied by 100) was calculated to determine the avidity index.

### 2.4. Statistical Analysis

GraphPad Prism Version 9 was used to compare means across populations using *t*-testing or ANOVA with Tukey’s post hoc adjustment for multiple comparisons as indicated. General linear models assessing the associations between ETEC antigens and immune responses as well as binary logistic regression were performed using SPSS v.27.

### 2.5. Ethics Statement

The serosurvey was approved by the Washington University Institutional Review Board (IRB, 202010227). Individual human serum collections were all obtained separately under approved protocols. Sharing of samples from Naval Medical Research Unit 6 was approved by the local IRB. Samples from Haiti were collected under works approved by the Washington University Institutional Review Board (202007027) and the National Bioethics Committee in Haiti (Comité National de Bioéthique). Samples from Egypt and Cameroon were collected under separate studies performed at Washington University.

## 3. Results

### 3.1. Seroprevalence

We hypothesized that humoral immune responses targeting ETEC antigens would be appreciably higher in individuals from endemic regions than in a nonendemic region (St. Louis, MO, USA). Using optimized ELISA assays, we explored the breadth of human responses to ETEC antigens from Cameroon, Egypt, Haiti, Peru, and the United States (Figure 1). Young children (<3 years old) in the United States would presumably be less likely to have traveled or be exposed to ETEC based on the low prevalence of ETEC in the US relative to adults [39,40]. Accordingly, we found that serum IgG responses to EatA, EtpA and LT were significantly elevated in Haitian children under three years of age relative to young children in the US (Figure 1A). In comparison to US adults, Cameroonian and Peruvian IgG responses to all three antigens were significantly elevated, while Egyptian subjects (≥10 years of age) had significantly elevated IgG responses to LT alone.

We also observed significant increases in IgA responses in children from Haiti and in adults. Young children in Haiti were found to have substantially elevated serum IgA responses to each of the antigens tested (Figure 1B) relative to US controls. Anti-EtpA responses were elevated in Peruvian adults relative to subjects in Egypt and Cameroon. However, US adults also had elevated IgA responses, which were significantly elevated compared to US children (*p* < 0.0001). For EatA, IgA responses were elevated in Peruvian adults relative to Egyptian subjects. US adult responses to EatA were again elevated, much like those observed in samples from Egypt. Finally, for LT, Peruvian IgA responses were also elevated relative to US adults. While IgA antibody levels in US adults were surprisingly higher than those of children in the US, we hypothesized that antibody avidity would be higher in individuals from endemic populations, reflecting more frequent exposures to ETEC. Intriguingly, however, adults from the United States (Saint Louis, MO, USA) were noted to have slightly higher IgA avidity to EtpA relative to individuals from Egypt or Peru. By contrast, anti-EatA IgA avidity values for Egyptian subjects proved to be higher than those observed in samples from Peru (Appendix A). Within individuals, antibody responses to each of the antigens (EtpA, EatA, and LT) and/or each isotype were variably correlated and these results differed by site (Appendix A). In general, however, EtpA responses tended to correlate with EatA responses, likely reflecting the conservation of these antigens and their common coexpression in individual isolates [21].

### 3.2. Immunogenicity

Antigens recognized during natural infections can focus vaccine design. Using data from a recently completed case–control study in Cap-Haïtien [37], we assessed immune responses to novel antigens following infection. We previously demonstrated that anti-EatA and anti-EtpA antibody responses increase with age and may correlate with protection from symptomatic disease in Bangladeshi children [24]. Similarly, we demonstrated that among Haitian children under three years of age, antibody responses increase with age (Appendix A) with the exception of anti-LT IgA responses. Conversely, changes in antibody responses over one month were negatively associated with age, suggesting children may reach maximal inducible responses as they age (Appendix A). In unadjusted analysis, EatA IgA and LT IgG were significantly induced upon antigen exposure relative to unexposed participants (Table 1).

In adjusting for additional variables associated with changes in immune responses, age remained significantly associated with changes in antibody levels (Table 2). Of note, the interaction effect between the EtpA and acute diarrheal symptoms was also significant, suggesting that symptomatic disease modifies the anti-EtpA IgG response.

Collectively, the data obtained in the present study reflect prior observations of immunogenicity of EtpA and EatA among children and adults in Bangladesh. The recent data lend further support to the idea that these antigens are widely distributed among ETEC and retain immunogenicity in diverse populations following natural ETEC infections.

## 4. Discussion

To further inform vaccine strategies to prevent ETEC infections, we characterized the immune response to candidate ETEC antigens and the LT toxin. We show that humoral immune responses to these antigens are common across a globally diverse set of samples. We selected sites that are either known or presumed to have high burdens of disease (Egypt, Cameroon, Peru, and Haiti) and compared our results to areas with a relatively low burden of disease (adults and children from the US). Our results confirm that ETEC infections in these populations are common and provide additional impetus to pursue ETEC vaccination strategies that include novel antigens EatA and EtpA, which have not been targeted in canonical approaches to vaccine development.

We utilized anti-LT responses as a marker of ETEC burden, as roughly two-thirds of all ETEC isolates express LT alone or in combination with ST. In endemic regions, anti-LT responses were universally elevated, supporting the commonality of ETEC exposure across these populations. While anti-LT responses cross-react with cholera toxin, there is limited reason to believe that our findings are reflective of cholera exposure, as cholera was only circulating in Cameroon at the time of sample collections (our study in Haiti pre-dates the reemergence of cholera that occurred in 2022). Moreover, anti-LT responses across a broad age range may suggest frequent reexposure, as immunity is thought to decline over time without exposure.

Our recent studies have focused on the potential role for EatA and EtpA to complement canonical approaches to ETEC vaccine development based primarily on colonization factor antigens. Antibodies directed at either novel antigen appear to limit intestinal colonization in animal models and impair pathogen–host interactions in vitro, a key step in the molecular pathogenesis of ETEC [41]. We have shown that these antigens are conserved among a diverse global collection of ETEC isolates [21,42], that they appear to contribute to virulence as well as adaptive immune responses to ETEC in young children in Bangladesh [24], and that they appear to be immunogenic in human volunteers [43,44]. However, the immunogenicity of these important virulence molecules has yet to be similarly demonstrated in diverse settings where ETEC remains highly endemic. Importantly, the present studies provide further evidence that these antigens are widely distributed among a diverse population of ETEC and that immunodominant regions of these molecules are retained. Our findings are also consistent with recent ETEC microarray data from studies on young children in Zambia conducted as part of phase I testing of the oral inactivated ETVAX vaccine that noted substantial IgG immunoreactivity to these antigens prior to vaccination [45].

Interestingly, we also surprisingly observed antibodies directed at these novel antigens in sera from US adults despite low titers in US children. This finding may reflect cross-reactivity with similar antigens from other species, travel to endemic regions, or exposure to ETEC within the US [39]. While ETEC have not been considered to be endemic in the US, multiple domestic foodborne outbreaks have occurred [46,47,48,49,50,51,52,53]. Because testing was previously not readily available in most clinical laboratories, sporadic cases of ETEC were frequently overlooked and only came to recognition following involvement of state or federal resources during outbreak investigations. Nevertheless, with the advent of culture-independent diagnostic testing platforms [54,55,56], ETEC have increasingly been identified in clinical laboratories. Indeed, recent studies indicate that less than half of domestic ETEC cases result from international travel, indicating the presence of as yet uncharacterized endemic foci of ETEC [40]. Previous molecular characterization of these domestically acquired isolates, including whole-genome sequencing, has demonstrated that they are remarkably similar to strains from known endemic regions, including expression of both EtpA and EatA [21].

The surprisingly high serum IgA levels directed against both EatA and EtpA among US adults relative to young children led us to examine whether IgA antibody avidity might differ from samples obtained in endemic regions. Although we presumed that more frequent exposure to ETEC in regions where it is endemic would result in more effective B cell maturation and higher antibody avidity to common ETEC antigens [57,58], this was not borne out in avidity testing, findings that could confound future vaccine studies in US adults. As antibody avidity is thought to reflect affinity of polyvalent antibodies in interaction with their respective target antigen, avidity has been used to discriminate between primary and secondary infections [59] and as a surrogate marker for protective immunity in vaccine trials [60]. High-avidity mucosal IgA can also facilitate clearance of pathogens from the intestinal lumen [61]. It is possible that differences in methodology for sample collection and preservation could have impacted the results of our comparisons. Nevertheless, further study is needed to replicate these findings and to explore responses to EtpA and EatA following vaccination.

Antigenic similarity between novel antigens studied here and virulence molecules of other pathogens could account for some of the immunoreactivity observed in serologic analysis involving US adults. EatA shares structural features, and likely shared epitopes with other autotransporter molecules including SepA of *Shigella* [62], with which it shares more than 70% identity [19]. Indeed, Eat or SepA has been identified in other *E. coli* pathovars [63,64], suggesting that exposure to these antigens may be relatively common. Likewise, EtpA shares overall structural homology with multiple antigens belonging to the two-partner secretion family of virulence molecules [65], including filamentous hemagglutinin, a component of acellular pertussis vaccines [66]. However, whether natural infections with *Bordetella pertussis* or vaccination elicits antibodies that cross-react with EtpA is presently unknown.

Elucidation of correlates of protection against ETEC infection would greatly facilitate vaccine efficacy studies [67,68,69]. Determination of fecal IgA, or enumeration of α4β7 integrin-expressing, “gut-homing,” antibody-secreting lymphocytes [70] may afford more direct mechanistic correlates of protection, yet neither approach is easily performed in field settings on large numbers of subjects. This has led to evaluation of serologic assays in evaluating protection afforded by vaccines for enteric pathogens [71]. Earlier seroepidemiologic studies of ETEC have yielded mixed results with respect to protection afforded preexisting serum antibodies against colonization factors or heat-labile toxin [7,72,73]. In earlier studies, we identified an association between anti-EatA or EtpA antibody responses and protection from subsequent symptomatic ETEC infection in Bangladeshi children [24]. Although we were able to demonstrate the immunogenicity of these antigens as Haitian children developed antibody responses after exposure to EatA and LT, as well as when symptomatic with EtpA+ ETEC infections, the work in Haiti was not designed to evaluate the protection associated with exposure to ETEC. Collectively, the results from Bangladesh and Haiti do suggest that ETEC expressing these antigens are likely circulating in diverse populations of young children where they are demonstrably immunogenic. Nevertheless, further data are needed to determine the protection afforded by immune responses to EatA and EtpA.

Although serosurveys can provide important insights regarding global disease burden and can be used to direct control or elimination strategies [74], they are prone to selection bias [75]. The present study is likely biased by the convenience of available samples and limited by acknowledged differences in sample collection and preservation.

Despite clear resource limitations in Haiti that pose significant risks for diarrheal diseases, there is currently surprisingly scant granular information regarding ETEC prevalence in this country. Large population-wide studies of antibody responses to multiple pathogens including ETEC (based on IgG antibodies to LT) suggest high transmission of these pathogens to young children early in life (<3 years of age) [32,76,77]. Similarly, molecular surveys conducted in Haitian cholera treatment centers and community-based oral rehydration units suggest that the burden of ETEC in young children is high [78]. ETEC have also been linked recently to growth limitations in a population of young children living in Cap-Haïtien, Haiti [79]. Here, we demonstrate significant age-dependent antibody responses to each of the ETEC antigens tested in young children, likely reflecting a large burden of illness, and intense transmission at an early age in this population.

Overall, the studies reported here also provide further evidence that EtpA and EatA, not presently targeted in ETEC vaccines under development, deserve further scrutiny as antigens that can complement or simplify an otherwise complex approach to achieving broad protection against these highly heterogeneous pathogens. Currently, ETVAX, a whole-cell killed vaccine of multiple strains targeting a total of four CFs (CFA/I, CS3, CS5, and CS6) combined with a novel toxoid (LTCBA) based on B subunits of LT and cholera toxin (CT), is the most advanced vaccine to enter later stages of clinical development [80,81,82]. Recombinant adhesins based on CFA/I [83] and CS6 [84] have also advanced through safety and immunogenicity studies. Although these canonical approaches may afford protection against related CF antigens, the remarkable diversity of these antigens as well as shifts in antigen prevalence over time [85] need to be considered in rational vaccine design [15]. Whether the addition of EtpA and EatA to existing strategies will obviate the need to target multiple CFs certainly needs to be tested empirically. However, it is hoped that additional details emerging from ongoing structure–function studies of these antigens combined with identification of critical protective epitopes recognized during ETEC infections will facilitate their inclusion toward development of a broadly protective vaccine.

## Figures and Tables

**Figure 1 microorganisms-11-02221-f001:**
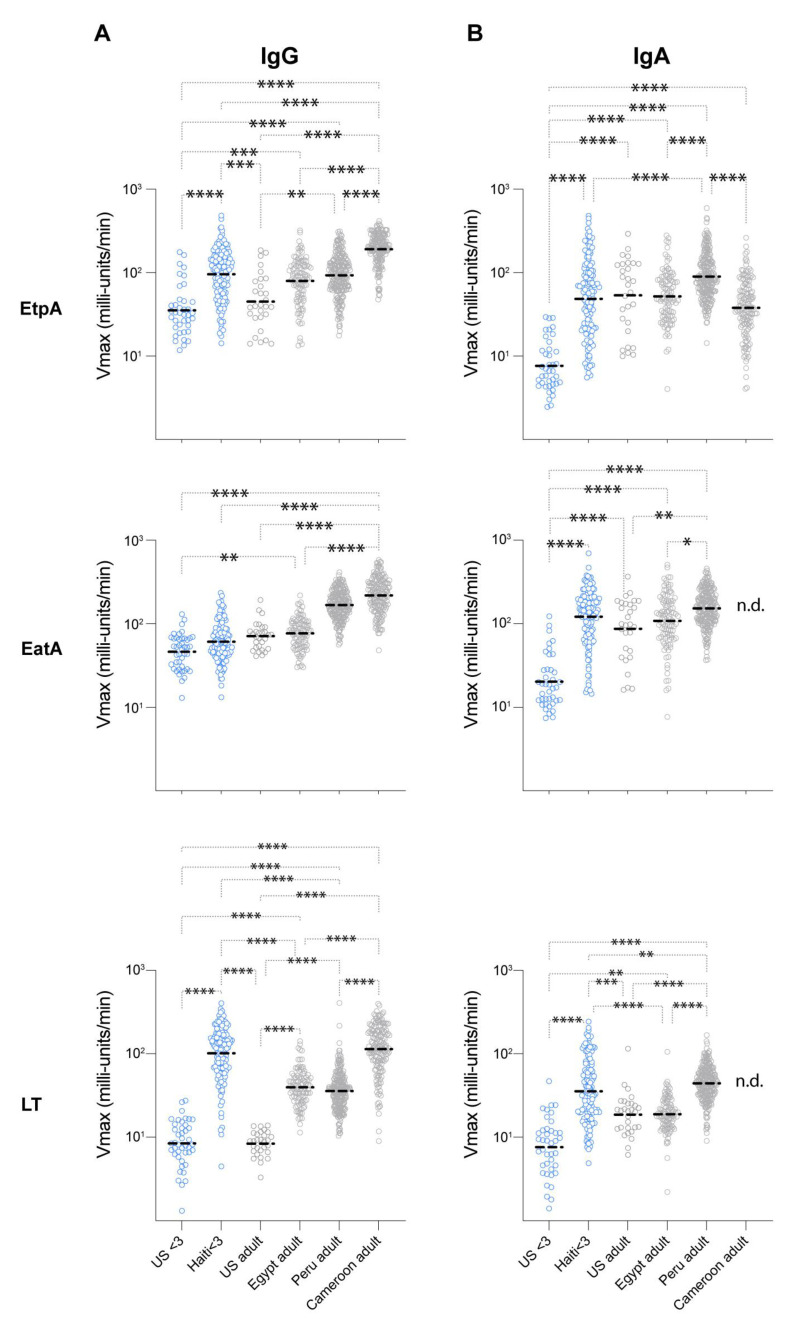
Immune responses to EtpA, EatA and LT in diverse populations. Shown are kinetic ELISA (IgG, (**A**)) data (IgA, (**B**)) with geometric mean values represented by dashed horizontal lines. *p* * < 0.05, ** < 0.01, *** < 0.001, **** < 0.0001 (Kruskal–Wallis nonparametric testing). n.d. no data due to insufficient quantity of sera for analysis.

**Table 1 microorganisms-11-02221-t001:** Unadjusted mean differences in antibody response one month after exposure.

Antigen	Isotype	Antigen				
		Present		Absent		
		Change *	N	Change	N	*p*-Value
EtpA	IgG	0.13	6	0.12	119	0.891
	IgA	0.04		0.14		0.689
EatA	IgG	0.08	6	−0.05	119	0.280
	IgA	0.32		0.01		0.034
LT	IgG	0.20	20	−0.02	105	0.006
	IgA	−0.02		−0.05		0.771

* Change is defined as the log_10_ (Vmax) response at follow-up subtracted from the baseline response.

**Table 2 microorganisms-11-02221-t002:** General linear model associating variables with immune response.

			Age	Symptoms	Antigen	Symptoms:Antigen *
EtpA IgG	0.074	*p*-value	0.008	0.081	0.945	0.045
		*η* ^2^	0.058	0.025	0.000	0.033
EtpA IgA	0.046	*p*-value	0.021	0.463	0.641	0.347
		*η* ^2^	0.043	0.004	0.002	0.007
EatA IgG ^†^	0.064	*p*-value	0.013	0.567	0.429	0.709
		*η* ^2^	0.051	0.003	0.005	0.001
EatA IgA	0.072	*p*-value	0.107	0.232	0.047	0.405
		*η* ^2^	0.021	0.012	0.033	0.006
LT IgG	0.223	*p*-value	0.010	0.001	0.001	0.494
		*η* ^2^	0.054	0.096	0.088	0.004
LT IgA ^†^	0.093	*p*-value	0.630	0.001	0.363	0.006
		*η* ^2^	0.002	0.081	0.007	0.062

* Significance of the interaction between symptoms and antigen immunogenicity. ^†^ Levene’s test of equality of variances is significant.

## Data Availability

The data presented in this study are openly available in, FigShare at the following doi: https://doi.org/10.6084/m9.figshare.24066372, https://doi.org/10.6084/m9.figshare.24066369, https://doi.org/10.6084/m9.figshare.24066366, https://doi.org/10.6084/m9.figshare.24066363, https://doi.org/10.6084/m9.figshare.24066360, https://doi.org/10.6084/m9.figshare.24066357 (accessed on 28 August 2023).

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
