# Peer review of "Seroprevalence Study of Conserved Enterotoxigenic Escherichia coli Antigens in Globally Diverse Populations"

_microorganisms, 2023, doi:10.3390/microorganisms11092221_

Round 1

Reviewer 1 Report

The study reports the results of a survey about the prevalence of antibodies targeting the protein EtpA and EatA, mainly expressed by ETEC.

The study is interesting, but, in my opinion, it shows two important limitations.

The first is the lack of data about cross-reactivity, which can lead to false positives. This has been generically addressed in the Discussion section, but without any kind of experimental investigation.

The second is the questionable quality of the language. Despite native English speakers being listed among the coauthors, it is far from acceptable. Following are listed only some of the issues: line 35: “remain” instead of “remains”; line 35: “are” instead of “is”; lines 41-42 “and as well as”; line 52: “ETEC share in their ability”; line 52: “ability” is not the most appropriate term. Please note that no more linguistic problems will be addressed in this review, but the entire manuscript has to be checked.

Following there are specific comments

Line 38: the sentence is not formally correct. ETEC is a group of organisms with a certain degree of heterogeneity, not highly diverse organisms. Why do they avoid adaptive immune selection? Because of their differences? The sentence is unclear.

Line 45: Please define LIMC.

Line 45 and elsewhere in the text: The study relies on data from endemic areas, but no detail is provided about which are the endemic areas.

Lines 44-46: please rewrite this sentence.

Lines 47-48: Why do ETEC infections drive empiric antibiotic use?

Lines 48-49: Why do natural infections appear to offer protection?

Lines 55-57: Why? The sense of this sentence is not clear. What class of proteins are the main targets?

Lines 68, 85, and 237: Why “inform”?

Lines 69-70: Do the Authors intend Enterobacteriaceae?

Line 73: As it is written (in italics), etpBAC is a gene, and the gene cannot secret anything.

Lines 86-87: This sentence is not clear.

Line 109: please specify the kind of sample.

Lines 175-181, 213-219: Those statements should be part of the Discussion, not Results, section.

Liners 181-206: The Authors define most of their data as significant but they do not show statistical results.

Discussion: The discussion is generally focused on the target, but some concepts are reiterated and redundantly expressed.

Please see the Comments and Suggestions for Authors

Author Response

We appreciate the efforts of the reviewer. While we did not concur with each statement, we have appended our detailed point-by-point response to each comment. 

Reviewer 2 Report

Your comprehensive study unveils intriguing patterns of immune responses to EtpA and EatA, raising thought-provoking questions about endemicity, antigenicity, and potential vaccine strategies.

Your methodology section may require minor revisions to address some errors.

Materials and Methods - Human Serum Samples:

Sentence 108: "Samples previously collected in Cameroon (2016)[32] or Egypt (Qalubiya Gover- 109 norate, circa 1997) ([33]) were obtained during the course of field studies on filarial infec- 110 tions." — Consider rephrasing the sentence to avoid confusion between the references and the dates.

Sentence 115: "This suspension was considered to be a 1:200 dilution (see supplemental methods)." — The text is clear, but you might specify the purpose of this suspension or why it was considered a dilution.

Line 118-119: — Consider rephrasing for clarity

Sentence 125: "Details of assay development and optimization are provided in supplemental information." — The text is clear, but consider rephrasing slightly for better flow: "Additional information regarding assay development and optimization is provided in the supplemental materials."

Line 138: “and stored, sealed, in the dark” — Rephrase for clarity: "dried for four hours at 37°C, sealed, and stored in the dark…….”

Few minor errors

Author Response

We greatly appreciate the efforts and assistance of reviewer 2. Our point-by-point response to the reviewer's comments is attached.  

Round 2

Reviewer 1 Report

The Author did not take into account most of the comments from this reviewer. In my opinion, language is not adequate, and most of the justifications are not adequate.

Then, the Authors wrote that the referee "offers no clear guidance regarding
further experimentation that could possibly address the perceived issue of “cross-reactivity". It is not a referee's task to design and set up experimental flow for the Authors, which should be able to develop a scientific strategy by themselves.

All those considering, I find that the indications have not been properly addressed, making this manuscript not suitable for publication.

In my opinion, the manuscript has not achieved a language level high enough to be published.

Author Response

The Author did not take into account most of the comments from this reviewer. In my opinion, language is not adequate, and most of the justifications are not adequate.

We wholeheartedly, but respectfully disagree with this reviewer's assessment of our manuscript. We have given each of this referee's criticisms serious consideration and made a number of changes that were previously suggested by this reviewer.  These changes were outlined in detail in the prior rebuttal to this reviewer's critique. 

Then, the Authors wrote that the referee "offers no clear guidance regarding further experimentation that could possibly address the perceived issue of “cross-reactivity". It is not a referee's task to design and set up experimental flow for the Authors, which should be able to develop a scientific strategy by themselves.

Again, we appear to be completely at odds with a referee that appears intent on raising issues without any clear justification for additional experimentation.  While we agree that it is not the role of the referee to design our experiments, the authors should not be tasked with chasing answers to spurious, vague and unsupported criticisms. We believe that we have sufficiently addressed the potential issue of cross reactivity with other antigens in the discussion. However, as noted in the previous rebuttal, there is no way to predict a priori what molecules might need to be examined to address this issue. Certainly, given the overall structural similarity EatA to passenger domains of dozens of other autotransporter molecules, or EtpA to other two-partner secretion exoproteins, the potential for cross reaction with conserved epitopes exists. Nevertheless, testing against a multitude of other potential cross reactive antigens is completely beyond the scope of the present study. 

All those considering, I find that the indications have not been properly addressed, making this manuscript not suitable for publication.

Obviously, we disagree with this referee's assessment. We believe that we had thoroughly addressed the referee's most substantive concerns in the previous revision.